# The application of resilience assessment grid in healthcare: A scoping review

**Mariam Safi**[ID][1,2,3]*, **Bettina Ravnborg Thude**[1,2], **Frans Brandt**[1,2], **Robyn Clay-Williams**[3]

**1** Internal Medicine Research Unit, University Hospital of Southern Denmark, Aabenraa, Denmark,
**2** Department of Regional Health Research, University of Southern Denmark, Odense, Denmark,
**3** Australian Institute of Healthcare Innovation, Macquarie University, Sydney, New South Wales, Australia

* Mariam.Safi2@rsyd.dk

## Abstract

**Data Availability Statement:** All relevant data are within the paper and its Supporting information files.

### Background

The Resilience Assessment Grid (RAG) has attracted increasing interest in recent healthcare discourse as an instrument to understand and measure the resilience performance of socio-technical systems. Despite its growing popularity in healthcare, its applicability and utility remain unclear. This scoping review aims to understand the practical application of RAG method and its outcomes in healthcare.

### Method

We followed the Arksey and O'Malley, and the Levac and colleagues' framework for scoping reviews and the PRISMA-ScR Checklist. We conducted searches of three electronic databases [Medline, Embase and Web of Science] in May 2021. Supplementary searches included Google Scholar, web and citation searches, and hand searches of the nine seminal edited books on Resilience Engineering and Resilient Health Care. All English language, empirical studies of RAG application in the healthcare setting were included. Open Science Framework [Registration-DOI. 10.17605/OSF.IO/GTCZ3].

### Results

Twelve studies met the inclusion criteria. Diversities were found across study designs and methodologies. Qualitative designs and literature reviews were most frequently used to develop the RAG and applied it in practice. Eight of the studies had qualitative designs, three studies had mixed-methods designs and one study had a quantitative design. All studies reported that the RAG was very helpful for understanding how frontline healthcare professionals manage the complexity of everyday work. While the studies gained insights from applying the RAG to analyze organizational resilience and identify areas for improvement, it was unclear how suggestions were implemented and how they contributed to quality improvement.

**Funding:** This work was supported by the University Hospital of Southern Denmark as part of a Ph.D. project. The funders had no role in study design, data collection and analysis, decision to publish, or preparation of the manuscript.

**Competing interests:** This work was supported by the University Hospital of Southern Denmark as part of a Ph.D. project. This does not alter our adherence to PLOS ONE policies on sharing data and materials.

## Conclusion

The RAG is a promising tool to manage some of the current and future challenges of the healthcare system. To realise the potential benefits of the RAG, it is important that we move beyond the development phase of the RAG tool and use it to guide implementation and management of quality initiatives.

## Introduction

The Resilience Assessment Grid (RAG) method developed by Hollnagel [1, 2] provides the conceptual basis for analyzing and supporting the organizational resilience of a complex system. The RAG proposes four abilities [responding, monitoring, learning and anticipating] that jointly enable resilient performance. Resilient performance means that an organization can function as required under expected and unexpected conditions alike such as changes, disturbances and opportunities [2]. Hollnagel suggests that it is not possible to directly measure resilience. However, the four abilities of RAG can help us understand what enables resilience performance in everyday work and conversely what, if it was missing, renders a system brittle [1, 2]. The intention with RAG is to use the four abilities as a set of proxy measures to construct a resilience profile of the system under assessment over time. The RAG provides a generic guideline as well as a set of example questions that need to be tailored for each new system to which it is applied [1, 2]. The set of example questions is provided alongside advice to develop *original* questions that are based on the underlying construct of the four resilience potentials, but tailored to fit the study setting.

Further, the RAG can be used to collect repeated measures on a single system, to inform continuous improvement and comparison. However, RAG is usually not applied to compare different systems or organizations, and is not considered by Hollnagel to be suitable for that purpose [3].

The RAG is based on resilience engineering (RE) principles [2, 4], which advocate for a system view that recognizes that adoption of a proactive approach is necessary for successful performance and safety in large complex socio-technical systems [5]. Given the challenges regarding safety, effectiveness and value shared by most healthcare systems [3, 4, 5], RE and RAG have garnered considerable interest. The RAG has been applied across several domains in addition to healthcare [6–8] to analyze and support resilience performance; these include aviation [9], air traffic management [10], nuclear power plants [11] and the water sector [12]. In healthcare, the RAG has been primarily applied in emergency care [7, 13] and anesthesia departments [14]. However, the RAG does not prescribe a specific method to develop the questions and rate them, which means that the RAG has been operationalized in different ways in different settings. Despite its growing popularity in healthcare, its applicability and utility remain unclear. There is no comprehensive description of the state-of-the-art application of the RAG in healthcare available to practitioners and academics who wish to be aware of its practical applications and developments since its inception. We, therefore, sought to understand the application of the RAG in healthcare to support resilient performance, including how and why it has (or has not) worked. To develop this knowledge, we undertook a scoping review of the literature on RAG applications in healthcare, focusing on empirical studies.

With a view to assisting practitioners and decision-makers as well as researchers, we set out to clarify:

- What methods are used to adapt the RAG?

- How has the RAG been put into practice and in which settings?

- What outcomes have been attributed to RAG application in healthcare?

## Methods

A scoping review methodology was chosen in order to examine the extent, range, and nature of research activity in the chosen field. We elected to use a scoping approach because it would allow for different study designs to be included and their findings synthesized. We followed the methodological stages outlined by Arksey and O'Malley [15] and further refined by Levac et. Al [16]. It entails: 1. Identifying the research question, 2. Identifying relevant studies, 3. Study selection, 4. Charting the data, and 5. Collating, summarizing, and reporting the results. This type of review does not include a quality assessment of the included articles. A protocol for the scoping review was published in the Open Science Framework (OSF)(Registration DOI. 10.17605/OSF.IO/GTCZ3). The reporting of the review is guided by the PRISMA-ScR Checklist [17] (S1 Appendix).

### Eligibility criteria

The inclusion criteria was to select studies from the healthcare setting, reporting on empirical research on RAG, published in peer-reviewed journals or books, as well as unpublished reports. These were restricted to English language. No limitations were set regarding publication year; however, the search was conducted in May 2021. Articles that described the RAG in non-healthcare settings, resilience papers not using the RAG, or describing individual or community resilience; conference abstracts with no full texts; and theoretical and review papers were all excluded.

We were not concerned with individual coping skills or resistance to stress and burnout but rather how the RAG method has been used in healthcare to understand how work system function.

Studies that applied the "four abilities of resilience engineering" were excluded, as the specific abilities identified were not necessarily consistent with those that comprised the RAG. We thereby avoided including studies that had a focus on resilience engineering theory, but did not apply the RAG method per se.

### Information sources

The scoping review aimed to identify and include a wide range of literature such as original research, book chapters, and reports, since much of the research on RAG is published in scholarly books and technical reports, rather than scientific journals. After the initial pilot testing of search terms, a final search strategy was developed with the assistance of a specialist librarian. The databases systematically searched were Medline, Embase and Web of Science. The search was developed for Medline and Embase and was translated to Web of Science. We included the term "health care or healthcare" in Web of Science since Web of Science also consists of non-healthcare sectors. This led to a balanced precision and sensitivity in the search. For this review, supplementary searches were necessary as well. Supplementary searches were conducted in Google Scholar, edited scholarly books on resilience engineering and a web search at the Resilience Health Care Net website. Additionally, we included the citations of the Resilience Assessment Grid paper [2] by Hollnagel as per advice of the specialist librarian. Table 1 shows the search strategy.

**Table 1. Search strategy.**

| | 
| --- |
| **Medline and Embase** |
| (resili* adj3 (grid or engineering or assessment or analysis)) |
| **Web of Science** |
| TS = ((resili* NEAR/3 (grid or engineering or assessment or analysis))) |
| AND |
| TS = (healthcare or health care) |
| **Google Scholar** |
| "resilience analysis grid" OR "resilience assessment grid" |

## Selection of sources of evidence

Reference details, including titles and abstracts where available, were downloaded into the reference management software Endnote X9 [18]. After removal of the duplicates in Endnote, all references were imported in Covidence [19]—a program that organizes and facilitates the initial screening of titles and abstracts and full-text screening. Two reviewers (RCW, MS), in parallel, independently applied the inclusion and exclusion criteria to title and abstract screening. In the next step, two reviewers (BRT, MS) independently assessed the full-texts of the included citations for final inclusion. A librarian assisted with retrieving the full-texts. In both abstract and full-text screening, the citations where the reviewers did not agree on inclusion were discussed until consensus was reached.

## Data charting process and data items

The Microsoft Excel data extraction spreadsheet was specifically designed for this review. Data concerning study characteristics, e.g., authors, year of publication, country where the study was conducted, aim, setting, and the methodology, e.g., design and data as well as information related to the objectives of the review, i.e., how RAG had been applied in healthcare and the outcomes of using RAG were extracted (Table 2). The data extraction form was piloted by one reviewer because of the small number of studies included in the review. The pilot resulted in minor changes to the extraction form. Three reviewers performed data extraction independently. The first reviewer (MS) extracted the data from all the studies (n = 12). The second (BRT) and third reviewer (RCW) divided the studies (50/50) randomly between them and extracted the data independently. Lastly, the extracted data was reviewed by the three reviewers, and disagreements were discussed in the group.

## Data processing and analysis

A narrative synthesis [20] was performed for this review. It described the scope of existing research and summarized data using structured narrative and summary tables. Data synthesis was undertaken in the following stages: i) evidence mapping ii) tabular and graphical presentation of the included studies.

The dataset was organized around the characteristics of the studies, content, and outcomes of RAG use in healthcare. The first author completed the analysis and conferred with the other authors regarding any uncertainties.

## Results

The systematic search retrieved 2229 citations, and after removal of duplicates, 1365 citations were retained. Overall, 42 references were considered potentially eligible. After the full-text

**Table 2. The characteristics of the included studies.**

| Author | Setting | Objective Participants | Publication type | Study design | Methods used to adapt the RAG | How was the RAG applied to study resilience performance |
|---|---|---|---|---|---|---|
| Alders, 2019 (UK) [7] | Acute Medical Unit (AMU) | **Objective** Investigating the resilient performance of AMU using RAG **Participants** Nurses (n = 77) | PhD thesis | PhD thesis Mixed-methods | • Observations • Focus group interviews (n = 18) **RAG questions** • 37 items: R(23)*, M(6)*, L(5)* & A(3)* • Original questions on deteriorating patients, allocations of workload, knowing what to do and when, knowing when colleagues need help, coordinating admissions and discharges. | • RAG survey (n = 77) • Follow-up semi-structured interviews (n = 6) • RAG repeated: No • RAG radar chart: Yes **RAG scoring** • 5-point Likert Scale (poor, fair, good, very good, excellent) participants rated the survey. • Calculated a mean score for each item and for each of the 4 resilience potentials. |
| Bertoni et al. 2021 (Brazil) [6] | ICU | **Objective** Investigating the social interactions of the RAG's four resilience abilities in an ICU using Social Network Analysis (SNA) **Participants** Doctors, nurses, nurse technicians, allied health professionals, psychologists, pharmacist, nutritionist, speech and occupational therapists (n = 133) | Journal article | Mixed-methods | • Not clear **RAG questions** • 13 items: R(1), M(1), L(1) & A(1) • Original questions on social interactions, frequency of the interactions, Interruptions, daily rounds, identifying peers/actors | • Survey (n = 133) • Individual semi-structured interviews (n = 5) • RAG repeated: No • RAG radar chart: N/A **RAG scoring** • 5-point Likert scale (never, less than once a month, one to three times a month, one to three times a week, daily) • Descriptive statistics and calculation of network metrics. |
| Chaung et al. 2020 (Taiwan) [25] | Hospital Emergency Department (ED) | **Objective** Developing the RAG and testing it in an ED **Participants** ED Director, Vice director, Physician, Head nurse (n = 4) | Journal article | Qualitative | • Translation of the original RAG into Chinese • Qualitative interviews (testing) to further improve the translated RAG. **RAG questions** • 37 items: R(10), M(7), L(11), A(7). Open-ended and close-ended. • Adapted the RAG questions to their system. | • Qualitative survey • 1 focus group interview (n = 4) • RAG repeated: No • RAG radar chart: Yes **RAG scoring** • 4-point Likert scale • 2 researchers coded and rated the answers for the qualitative questions. • Calculated the average score for each potential and presented it in percentages. |

*(Continued)*

**Table 2.** (Continued)

| Author | Setting | Objective Participants | Publication type | Study design | Methods used to adapt the RAG | How was the RAG applied to study resilience performance |
|---|---|---|---|---|---|---|
| Chuang et al. 2020 (Taiwan) [13] | Hospitals, 4 EDs | **Objective** Resilience performance of emergency departments (EDs) **Participants** Director, Administrative staff, Physician, Nurse (n = 16) | Journal article | Qualitative | • Qualitative RAG survey. Chuang et al. 2020 [25] <br><br>**RAG questions** Chuang et al. 2020 [25] | • Qualitative survey <br>• Focus group interviews (n = 4) <br>• RAG repeated: No <br>• RAG radar chart: Yes <br><br>**RAG scoring** Chuang et al. 2020 [25] |
| Clay-Williams & Braithwaite 2019 (Australia) [26] | Hospital, Emergency services | **Objective** RAG was used to look at the successes and failures of information systems when challenged by the thunderstorm asthma **Participants** N/A | Journal article | Qualitative (Document analysis) | N/A <br>**RAG questions** N/A | • Document analysis <br>• RAG repeated: N/A <br>• RAG radar chart: N/A <br><br>**RAG scoring** N/A |
| Darrow 2017 (US) [21] | 2 Primary care locations 1 ED | **Objective** Assessing organizational and individual resilience using RAG **Participants** Physicians, residents, medical assistants, nurses, receptionists & management employees. (n = 12) | Graduate thesis | Qualitative | • RAG interview guideline <br>• Literature review <br>• Lindvall et al. [28]. <br><br>**RAG questions** <br>• 5 categories of resilience <br>• 54 items (34 items on 4 resilience potential and 16 items individual resilience) <br>• Original questions on: resourcefulness, communication, sensemaking and bricolage, team efficacy, safety, psychological safety, problem solving | • Qualitative survey (n = 12) <br>• RAG repeated: No <br>• RAG radar chart: Yes <br><br>**RAG scoring** <br>• 7-point Likert Scale (strongly disagree, disagree, somewhat disagree, neither agree nor disagree, somewhat agree, agree, strongly agree) <br>• 3 researchers analyzed the interview data and rated the items. <br>• The three raters' ratings on the five categories of resilience were then averaged. |
| Falegnami et al. 2018 (Multi-country survey. 16 nations) [14] | Anesthesia department | **Objective** Measuring the resilience performance of anaesthesiologists **Participants** Anaesthesiologists (n = 172) | Journal article | Quantitative (Cross-sectional study) | • Literature review <br>• Analytic hierarchy process (AHP) framework <br><br>**RAG questions** <br>• 57 items: R(8), M(6), L(8) & A(7). <br>• Original questions on communication, team work, role and responsibilities, expertise, evaluation of process. | • RAG survey (n = 172) <br>• RAG repeated: No <br>• RAG radar chart: Yes <br><br>**RAG scoring** <br>• 5-point Likert scale. <br>• Different rating score depending on the question. <br>• Aggregated resilience for the 4 potentials were computed using the AHP calculations. |

(*Continued*)

**Table 2.** (Continued)

| Author | Setting | Objective Participants | Publication type | Study design | Methods used to adapt the RAG | How was the RAG applied to study resilience performance |
|---|---|---|---|---|---|---|
| Hegde et al. 2020 (US) [23] | Hospital, multi-specialty including emergency medicine, surgery, urogynecology, radiology, orthopedics, and internal medicine | **Objective** To understand how frontline healthcare providers achieve safe and effective patient care in their everyday work **Participants** Frontline clinicians, senior safety administrator, participants with leadership and decision making roles (n = 18) | Journal article | Qualitative | • RAG used in the interview protocol.<br><br>**RAG questions** N/A | • Interviews<br>• RAG used in analysis of interview data<br>• RAG repeated: N/A<br>• RAG radar chart: N/A<br><br>**RAG scoring** N/A |
| Hegde et al. 2014 (US) [24] | Hospital, multi-specialty including emergency medicine, surgery, urology, orthopedics, critical care and internal medicine | **Objective** Understand resilience in healthcare organization through the lens of frontline health care workers **Participants** Physicians, residents, nurses. (n = 14) | Journal article | Qualitative | • RAG used in the interview protocol.<br><br>**RAG questions** N/A | • Interviews<br>• RAG used in analysis of interview data<br>• RAG repeated: N/A<br>• RAG radar chart: N/A<br><br>**RAG scoring** N/A |
| Hunte & Marsden 2019 (Canada) [27] | Hospital, ED | **Objective** Resilient performance of an urban emergency department using RAG **Participants** Participants with leadership, clinical, clerical, technical, educational and organizational roles. (n = 35) | | Qualitative | • Workshops<br><br>**RAG questions** • 23 items: R(6), M(6), L(6) & A(5).<br>• Adapted the RAG questions | • Qualitative survey<br>• RAG repeated: Yes<br>• RAG radar chart: Yes<br><br>**RAG scoring** • Not clear |
| Mahmoud et al. 2020 (Australia) [8] | Hospital, Operating theatre, operating rooms | **Objective** Explore the resilience characteristics of Operating Theatre (OT) **Participants** OT nurses (n = 28) | Book Chapter | Qualitative | N/A **RAG questions** N/A | • RAG framework used in the analysis of the interview data<br>• RAG repeated: N/A<br>• RAG radar chart: N/A<br><br>**RAG scoring** N/A |
| Patriarca et al. 2018 (Italy) [22] | Hospital, Anesthesia department | **Objective** Resilience in sociotechnical system to support decision making from a safety management perspective **Participants** Neuro- anaesthesiologists (n = 12) | Journal article | Mixed-methods | • Literature review<br>• Focus group interview (n = 3)<br>• Analytic hierarchy process (AHP) framework<br><br>**RAG questions** • 57 items: R(8), M(6), L(8) & A(7).<br>• Original questions on communication, team work, role and responsibilities, expertise, evaluation of process. | • RAG survey<br>• Follow-up focus group interview<br>• RAG repeated: No<br>• RAG radar chart: Yes<br><br>**RAG scoring** • 5-point Likert scale.<br>• Different rating score depending on the question.<br>• Aggregated resilience for the 4 potentials were computed using the AHP calculations. |

*Respond (R), Monitor (M), Learn (L) and Anticipate(A)

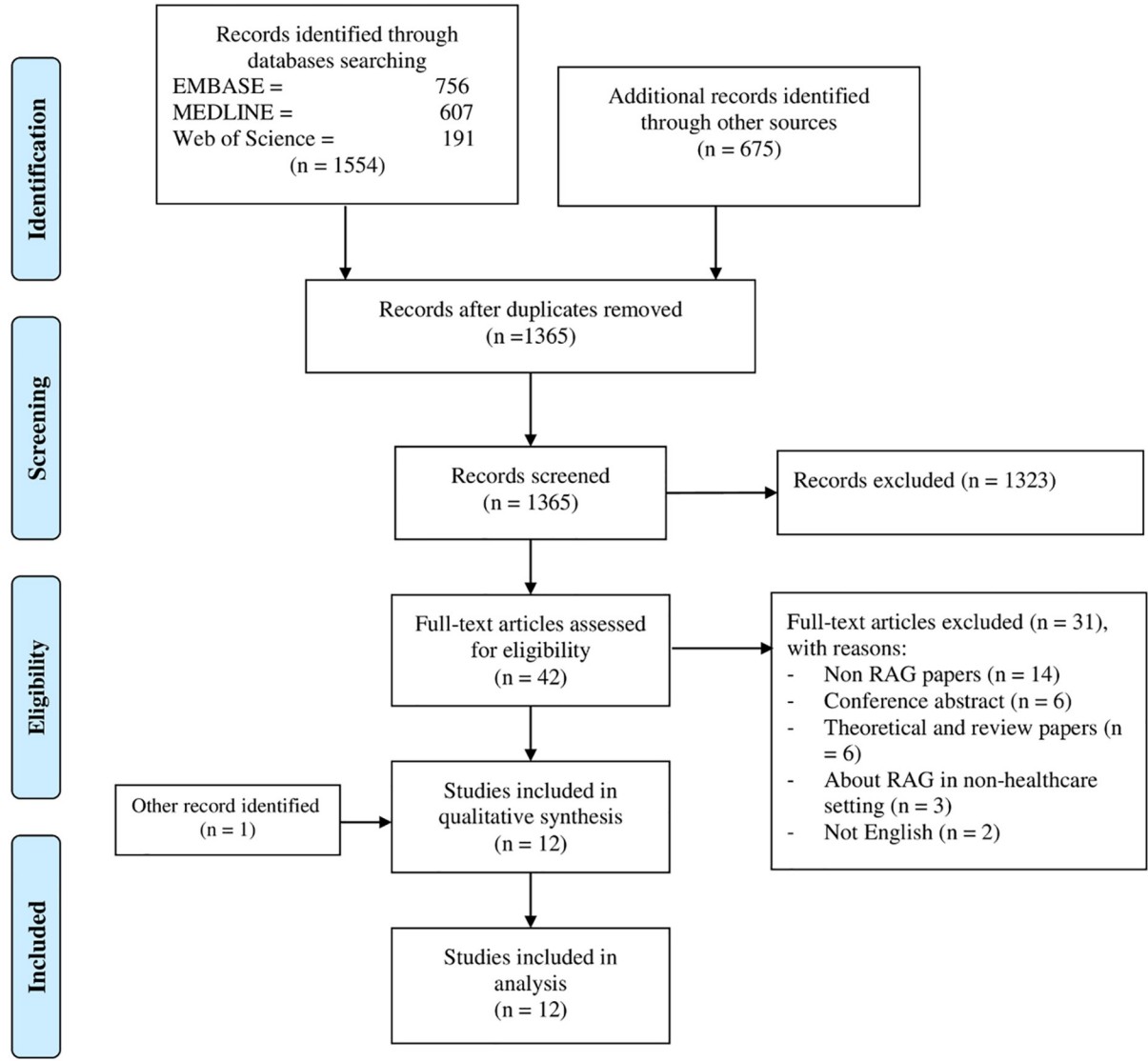

**Fig 1. PRISMA flow diagram.**

assessment, 12 were included and 31 articles were excluded. Fig 1 illustrates the inclusion and exclusion of citations at each stage of the screening process, using the PRISMA flow diagram.

## Characteristics of the studies

Table 2 provides overview of the characteristics of the included studies. The included studies originated from a wide range of countries such as Taiwan, Australia, Canada, United Kingdom, Brazil and United States (see Fig 2). More than 50% of the studies were peer reviewed journal articles. Fig 3 shows the distribution of included reviews published per year (2014–2021) and by document type. Fig 3 shows a small increase in peer reviewed publications from 2018 onwards. The studies were primarily conducted in hospital settings. Only one study had included two primary care locations [21]. The majority of the studies about RAG were conducted in hospital emergency and anesthesia departments [14, 22]. The remainder were single studies in multi-specialty hospitals [23, 24], in intensive care unit [6] and operating theaters

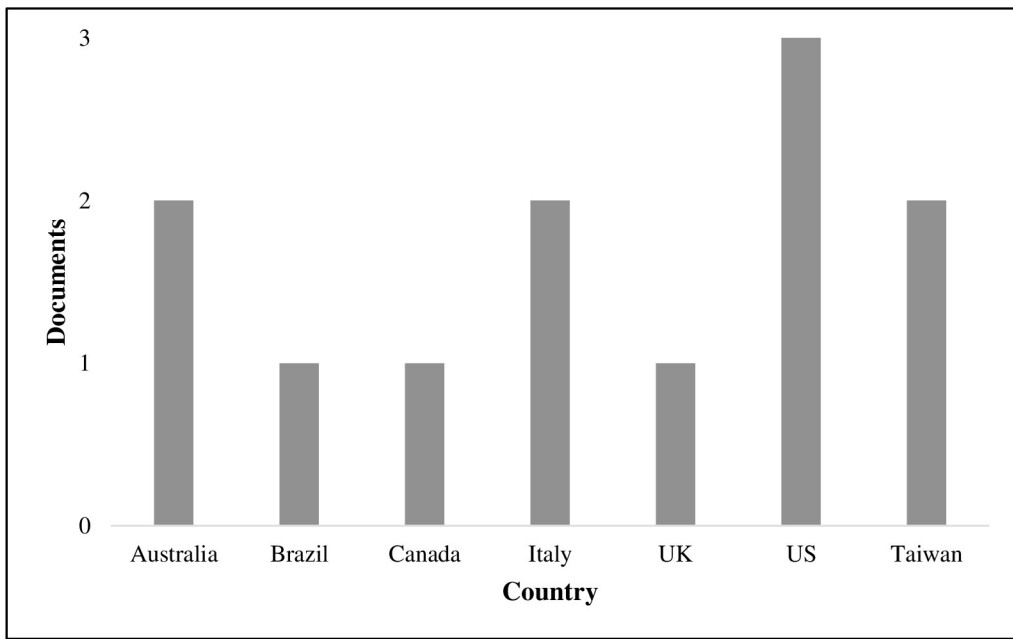

**Fig 2. Distribution of published reviews by country.**

[8]. Eight studies [8, 13, 21, 23–27] employed a qualitative design, three studies [6, 7, 22] a mixed-methods design and one study [14] was purely quantitative (see Fig 4). All studies employed a case study design except one study [14] which had cross-sectional design.

The primary purpose of the included studies was how to develop or adapt the RAG in order to assess or understand the resilience performance of the selected study setting or organization.

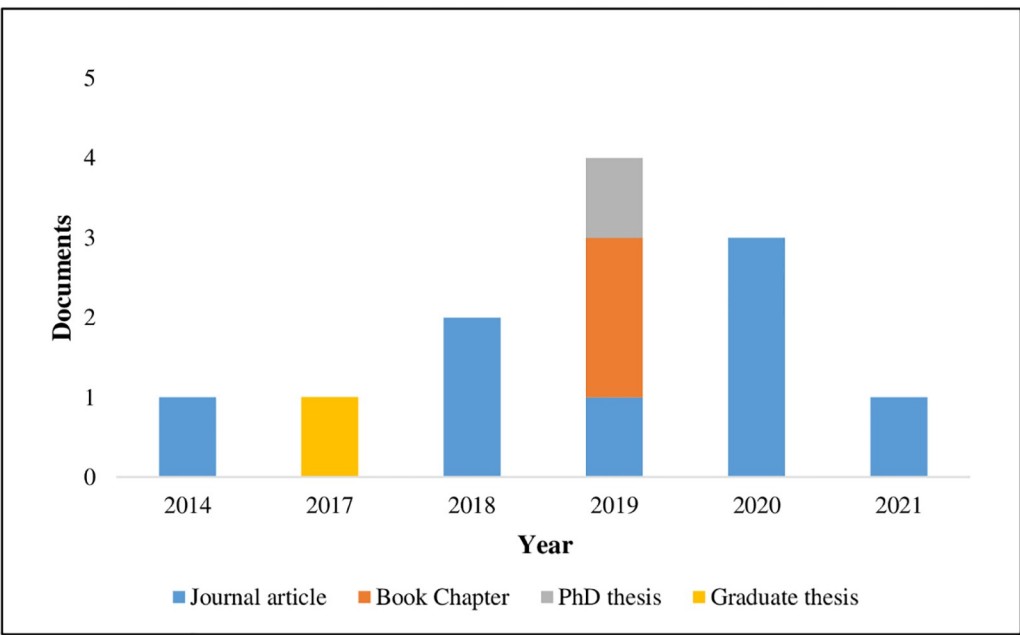

**Fig 3. Distribution of the published reviews over time and by study type.**

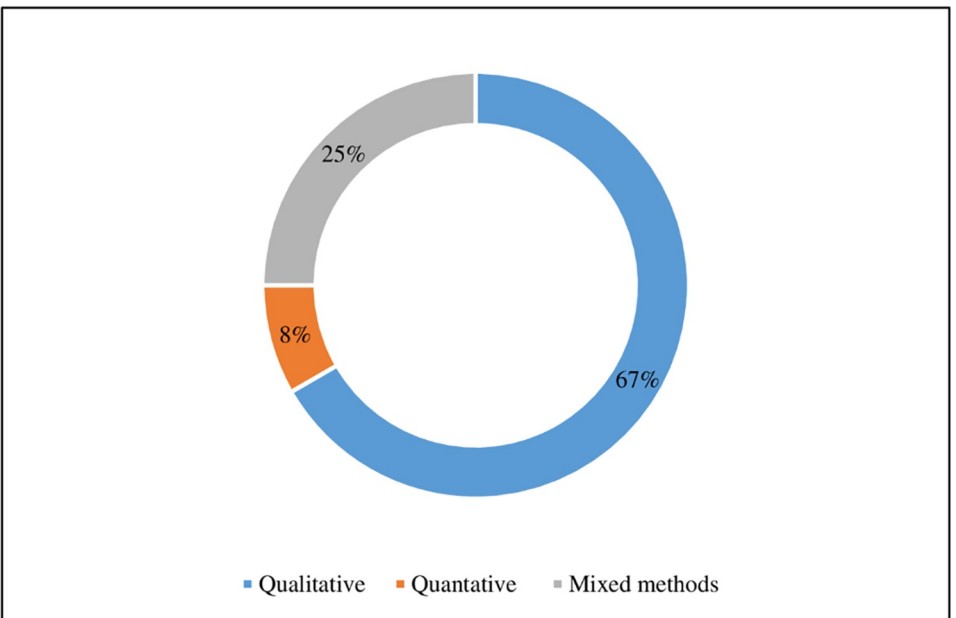

**Fig 4. Study designs of the included studies.**

Eleven studies were prospective. The remaining single study [26] used the RAG to retrospectively examine the success and failures of information systems that were challenged by an unexpected event, thunderstorm asthma.

## Ways of contextualizing the RAG

In the included studies, qualitative methods and literature review were most commonly used to adapt and develop the RAG tool for the specific study setting (number of the studies). In the study by Alder [7], focus group interviews and individual semi structured interviews were conducted based on the RAG with the nurses in the ICU. The interviews were used to generate the RAG questionnaire referring to specific system activities such as deteriorating patients, allocation of workload, knowing what to do and when, knowing when colleagues need help, and coordinating admissions and discharges. Hunte and Marsden [27] used dialogue workshops consisting of frontline care providers, support staff and leaders to explore and refine the RAG components. The content of the ED specific RAG was derived from these workshops. In the study by Patriarca et al. [22] the generic RAG questions were adapted to the anesthesia domain based on a literature review. Following this, the Analytical Hierarchy Process (AHP) and focus group interviews were used to refine the AHP-RAG. AHP is a method used for organizing and analyzing complex decisions using pairwise comparison. Patriarca et al. [22] used AHP to define the weights of the items. Questions on communication, teamwork, role and responsibilities, expertise, evaluation of process etc. were developed. The study by Falegnam et al. [14] further refined the original AHP-RAG by Patricia et al. to fit the anesthesia domain. This was conducted with the initial literature review and a panel of three subject matter experts (SMEs), who revised the AHP-RAG questionnaire to describe properly the four RAG resilience potential in terms of items related to the anesthesia domain. Likewise, Darrow et al. [21] adapted the original RAG questions to fit the system with the assistance of a literature review and the resilience survey by Lindvall et al. [28]. However, Darrow had included a fifth resilience category, i.e. individual resilience. The questionnaire under each resilience category referred to

questions such as resourcefulness, communication, sensemaking and bricolage, team efficacy, safety, psychological safety and problem-solving. The two studies by Chuang et al. [13, 25] are related. In the first study [25] the original RAG questionnaire was first translated into the Chinese language and then edited to fit the corresponding ED system under study by an expert panel. Focus group interviews were used to test and further refine the ED-RAG. In the second study [13], the ED-RAG had some minor changes and was applied in 4 ED hospitals. Bertoni et al. [6] used the RAG framework in combination with Social Network Analysis (SNA) to model interaction between the four resilience abilities in an ICU. The SNA-RAG included questions on social interactions, frequency of the interactions, interruptions, daily rounds, and identifying peers/actors. The remaining studies [8, 23, 24, 26] used the RAG as a framework to develop interview guides and for data analysis to understand resilience.

Additionally, in the studies [7, 13, 14, 22, 25, 27] there was some imbalance in the number of items for each of the four resilience potentials. For example in the study by Alders [7], the potential to respond had 23-items, monitoring had 6-items, learning had 5-items and anticipating had 3-items. In the study by Darrow [21], the distribution of the items for each of the four resilience potentials was not clear.

## Ways of applying the RAG

In the included studies, qualitative methods were most commonly used to put the RAG tool into practice. Of these studies, two involved focus group interviews [13, 25], one involved semi-structured interviews [21], one used the RAG in document analysis [26], one study applied the RAG in monthly meetings [27] and three studies [8, 23, 24] used the RAG framework to analyze qualitative data retrieved through interviews and observations. In the qualitative studies, there was a very large sample size variation. The minimum sample size for each setting was minimum 4 and maximum 35. The interviews included a wide range of professional groups, including physicians, nurses, receptionists, and employees with leadership and management roles at various levels. However, the majority of the respondents were frontline healthcare clinicians.

The quantitative studies included four studies [6, 7, 14, 22] that applied the RAG tool in survey format. Three of the studies [6, 7, 22] conducted qualitative interviews to elaborate on the survey results. One of the study [6] used the RAG method in combination with social networking theory to shed light on the ICU actors interactions in relation to the four resilience potentials. The survey participants were physicians, nurse technicians, allied health professionals, psychologists, pharmacists, nutritionists, and speech and occupational therapists. The study by Alders [7] sent a RAG survey to nurses at different levels in an Acute Medical Unit. The study calculated the mean for each item and the overall mean for each resilience potential. The two remaining studies [14, 22] were conducted in the anesthesia department. The first study [22] applied the AHP-RAG survey and involved 12 neuro-anesthesiologist. The second study [14] adopted the original AHP-RAG and applied it in a multi-country survey involving anesthesia professionals to test and validate the AHP-RAG survey. Two [14, 22] out of four studies visualized the results in radar chart or star plot as well. The Radar charts are a way to visualize multivariate data and make visible concentrations of strengths and weaknesses. The radar chart is not itself a measure of resilience [4].

Furthermore, the studies used different methods to define the answer's score against each question. Most of the studies [6, 7, 13, 14, 21, 22, 25] used a 4-7-point Likert scale. In some of the studies [13, 21, 25], the researchers coded and rated the answers. Hence, the calculation of the average score for each item and for each of the four resilience potentials differed, see Table 2.

## Outcomes of resilience assessment grid use in healthcare

All the included studies reported that the RAG was very helpful for understanding how front-line healthcare professionals manage the complexity of everyday work. Following the application of RAG, the studies most commonly reported; increased staff engagement through dialogues and discussions related to the four resilient potentials; identified areas for quality improvement; provided an overview about how well the system responds to expected and unexpected conditions in daily operations; assisted with improving the patient pathway; assisted with improving working condition; and provided a resilient profile of the system or organization related to response, monitor, learn and anticipate(see Table 3.).

Although 12 studies identified areas for improvement, only two studies reported that the improvements had been implemented. In the study by Hunte & Marsden [27] the RAG was used in an emergency department to enhance their collective ability to manage complexity in everyday practice and monitor their work overtime. Following the development of the context specific RAG, the emergency department: i) initiated a monthly interdisciplinary departmental meeting to consider each domain of the RAG; ii) implemented operational metrics to assess their performance, and iii) created an action plan linked to the metrics. Additionally, out of the 12 studies, only the Hunte & Marsden study repeated the RAG. In the study by Hegde et al., a Resilience Mapping Framework [RMF] [23] was developed to illustrate the resilience capabilities and their relationships across different levels of organizational scale for specific cases. Following this, the study applied the RMF to specific issues, such as how to manage patients that have the propensity to become violent and reviewing patients charts before surgery to detect potential harm.

The outputs and benefits identified in the studies were not systematically measured. While the findings showed that the staff identified system level interventions, it was unclear how the suggestions should be implemented and how they added to quality improvement.

## Discussion

In this review we aimed to provide an overview and synthesis of how the RAG method has been applied in healthcare. The review found only 12 studies that had applied RAG in the healthcare setting. The RAG method was primarily applied in the hospital setting. The included studies had primarily qualitative and mixed-method designs. While there was diversity across study design and methods, qualitative methods and literature reviews were most frequently used to develop the RAG and to apply it in practice. Only four of the studies [6, 7, 14, 22] used quantitative surveys to assess resilience performance. Three of these four [6, 7, 22] studies also conducted qualitative interviews to further elaborate the quantitative results. Furthermore, some of the studies [8, 23, 24, 26] applied the RAG as a framework for data analysis

**Table 3. Outcomes of resilience assessment grid use in healthcare.**

| Outcomes attributed to RAG | Articles |
|---|---|
| Understanding the resilient system performance in relation to responding, monitoring, learning and anticipating. A resilience profile of the system | [6–8, 13, 14, 21–27] |
| Increased staff engagement. | [7, 27] |
| Identified areas for quality improvement. | [6–8, 13, 14, 21–27] |
| Provided an overview of everyday operational work. | [6–8, 13, 14, 21–27] |
| Assisted with improving patient pathway and thereby the working condition. | [27] |
| Provided an overview of how the system responds to unexpected events. | [26] |

to understand resilience in relation to the four resilience potentials. Once a RAG is performed, it can be repeated to monitor the continuous improvement of the system or organization over time [2]. While this is the intended purpose of the instrument [1, 2], only one study [27] reported that the RAG was repeated in monthly interdisciplinary meetings to monitor the ED's performance and work in practice. This may be impacted by financial and other constraints on the project, such as funding and time. A solution to this can be to apply the RAG pre- and post-organizational changes, or to update the survey items and repeat it again.

While the included studies were heterogeneous in terms of data collection, the review showed that the RAG was frequently applied in high-risk areas within healthcare, such as the acute care and the anesthesia domains. The majority of the studies demonstrated how a RAG questionnaire can be developed and used to profile the performance of an organization in terms of the four resilience abilities and identify areas for improvement. Similarly, the RAG tool was applied in other industries such as the oil and gas industry [29], the water sector [12], the Swedish Civil Aviation Administration [9], and process industries [30]. The wide areas of application of the RAG method demonstrates its flexibility and applicability in different types of complex socio-technical systems and at different scales. However, both in the healthcare sector and in other industries there is an implementation gap. It is unclear how the results from RAG contributed to quality improvement initiatives that were put into practice. The suggestions for quality improvement in the studies required further work.

Another concern, that has been raised regarding the RAG method is that it does not prescribe weighting the four abilities. The review confirmed this problem, finding imbalance in the number of items for each of the four resilience potentials in the studies. Hollnagel [2] reiterates the importance of addressing all the abilities to some extent to achieve resilience. Apneseth et al. [29] argues that the relative importance of the four abilities of RAG varies depending on the system in question. While all organizations to some extent need to develop the ability to respond, many organizations also put some effort into the learning ability to improve their responses to activities. However, not all organizations needs to put the same emphasis on monitoring and anticipating ability in order to be resilient [29]. Another option may be to use the analytical hierarchy process framework proposed by Patriarca et al. [22] to study the different weights of the abilities and their relative influence on overall organizational resilience. However, the AHP is a time-consuming and difficult process [22], which may discourage practitioners and managers from using the RAG framework.

Additionally, the review showed that most of the studies used the underlying construct of the RAG to develop the questions pertinent to the 4 resilience potentials, i.e. respond, monitor, learn and anticipate. In the RAG surveys, each of the four resilience potentials included subdimensions or themes such as communication, sensemaking, role and responsibility, teamwork, expertise, social interaction, and interruptions. Only three of the studies [13, 25, 27] followed the original RAG questions closely and edited them to fit their system. The study by Darrow [21] included a fifth category, i.e. individual resilience. This indicates the RAG's flexibility and its increasing potential benefits to cover all aspects of the sociotechnical system. The review also showed that there is no clear method dictating how to define and rate the answer's score against each question. Hence, some researchers calculated the average score for each item and for each of the four resilience potentials, see Table 2. Aggregating items in this way is a problematic approach, if not accompanied by a statistical analysis (such as a factor analysis) to show that the aggregated items are related.

Resilience in itself is considered to be difficult or impossible to measure. Some other methods and tools such as the Functional Resonance Analyses Method (FRAM) method [31–35], Concepts for Applying Resilience Engineering (CARE) model and the Team Resilience Framework(RMF) [36–38], have also been applied in the healthcare setting to elucidate the

complexity of everyday clinical work and to suggest new perspective to improve safety and resilient performance in daily routines. Most of the methods used in healthcare to understand resilient performance are still in their early development and are more theoretical rather than ready-made tools for practical use in organizations [39]. The RAG and FRAM are conceptually clear and established methods [7] that have been used across different healthcare settings. Compared to other tools and methods, the RAG provides healthcare organizations sufficient methodological guide for how to assess, analyze and manage resilient performance at system level [1, 2, 7].

This review showed that the RAG is frequently applied in practice by other researchers to understand the resilient performance of their system. However, the literature shows that the application of the RAG tool has not moved beyond the developing and testing phase, to using the RAG tool to implement and manage quality initiatives in healthcare. The application of the RAG method in healthcare is still evolving, evidenced by the small increase in peer-reviewed publications since 2018. Most of the research on RAG is conducted in developed countries; however, more research is needed to investigate whether the RAG is applicable in countries with different health systems or more limited resources. Future studies should focus on how to use the knowledge derived from application of the RAG to implement changes in the system leading to improved outcomes.

## Strengths and limitations

The review followed a comprehensive methodology for scoping reviews as outlined by Arksey and O'Malley [15] and the reporting of the study followed the PRSIMA-ScR [17]. A research librarian was also involved in the review process and assisted with developing the search strategy and retrieving the full texts, which strengthened the quality of our review. To our knowledge, this is the first scoping review on the RAG and its application in the healthcare setting.

Although only English language articles were included in the review, some of these reported on studies where the RAG was applied in languages other than English.

## Conclusion

Research on the RAG began approximately a decade ago. It is a young and evolving method and has been applied in different industries, including healthcare, to investigate organizational resilience. The results from the RAG can enhance understanding of everyday operational work, identify challenges to safe and effective patient care, and create a resilience profile of the system or organization related to four resilience potentials (respond, monitor, learn, anticipate). While the reviewed studies gained insights from applying the RAG to analyze organizational resilience, it was unclear in many cases if improvements had been implemented. Despite the fact that the RAG provides a tool for managers to support the resilience performance of a complex socio-technical system, most studies reported that application was limited to a specific unit or department without the strong support of the senior managers, which led to limited organizational reach. To realize the potential benefits of the RAG, it needs to be accessible and manageable when translated to a new setting. More research is needed to investigate how the RAG method can be used to not only identify quality improvements but to implement and manage them in practice. Further, it is important that in the process of developing and applying the RAG, senior managers and other key stakeholders are involved in order to ensure that identified improvements are implemented, and evaluated through repeating the RAG post-intervention.

## Supporting information

**S1 Appendix. PRISMA-ScR checklist.**
(DOCX)

## Author Contributions

**Conceptualization:** Mariam Safi, Bettina Ravnborg Thude, Frans Brandt, Robyn Clay-Williams.

**Data curation:** Mariam Safi, Bettina Ravnborg Thude, Robyn Clay-Williams.

**Formal analysis:** Mariam Safi.

**Funding acquisition:** Mariam Safi, Frans Brandt.

**Investigation:** Mariam Safi, Bettina Ravnborg Thude, Frans Brandt, Robyn Clay-Williams.

**Methodology:** Mariam Safi, Bettina Ravnborg Thude, Robyn Clay-Williams.

**Supervision:** Robyn Clay-Williams.

**Validation:** Mariam Safi, Bettina Ravnborg Thude, Robyn Clay-Williams.

**Visualization:** Mariam Safi.

**Writing – original draft:** Mariam Safi.

**Writing – review & editing:** Bettina Ravnborg Thude, Frans Brandt, Robyn Clay-Williams.

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
