## [Decision Letter · Decision Letter 0]

30 May 2022

PONE-D-22-05861The application of resilience assessment grid in healthcare: A scoping reviewPLOS ONE

Dear Dr. Mariam Safi

Thank you for submitting your manuscript to PLOS ONE. After careful consideration, we feel that it has merit but does not fully meet PLOS ONE’s publication criteria as it currently stands. Therefore, we invite you to submit a revised version of the manuscript that addresses the points raised during the review process.

We look forward to receiving your revised manuscript.

Kind regards,

Sonu Goel, MD

Academic Editor

PLOS ONE

Journal Requirements:

This work was supported by the University Hospital of Southern Denmark as part of a Ph.D. project. 

This work was supported by the University Hospital of Southern Denmark as part of a Ph.D. project. 

This work was supported by the University Hospital of Southern Denmark as part of a Ph.D. project. 

6. Thank you for stating the following in the Competing Interests section: 

This work was supported by the University Hospital of Southern Denmark as part of a Ph.D. project. 

7. We note that you have referenced (ie. Bewick et al. [5]) which has currently not yet been accepted for publication. Please remove this from your References and amend this to state in the body of your manuscript: (ie “Bewick et al. [Unpublished]”) as detailed online in our guide for authors

8. Please include a copy of Table 3 which you refer to in your text on page 12.

Additional Editor Comments:

Dear Authors.

Thank you for submitting your contribution to the journal.

We have received reviews on your manuscript from our reviewers.

Kindly address the comments and upload the revised manuscript.

Thank you

Reviewers' comments:

Reviewer's Responses to Questions

**Comments to the Author**

1. Is the manuscript technically sound, and do the data support the conclusions?

Reviewer #1: Partly

Reviewer #2: Yes

2. Has the statistical analysis been performed appropriately and rigorously? 

Reviewer #1: Yes

Reviewer #2: N/A

3. Have the authors made all data underlying the findings in their manuscript fully available?

Reviewer #1: Yes

Reviewer #2: Yes

4. Is the manuscript presented in an intelligible fashion and written in standard English?

Reviewer #1: Yes

Reviewer #2: Yes

5. Review Comments to the Author

Reviewer #1: Article looks interesting.

Over all comment- spacing issues with respect to putting references in the text(numbers and et.al)

line 33- start the sentence with 12 in words not number.

line 68-reference missing

line 102-May 21 needs to be in sentence, to make more sense.

line 242-it should be- actors

line 247- it should be- conducted in the anesthesia area

line 282- omit- otherwise

line 343- replace is with are

Reviewer #2: A methodologically sound review paper.

The authors have addressed an important issue in appraising resilience in health care settings.

Like most issues which are important, resilience is difficulty to quantify. It will also be influenced by different settings and cultures.

The present study can be considered very preliminary and the concepts need to be validated in different settings.

6. PLOS authors have the option to publish the peer review history of their article (what does this mean?). If published, this will include your full peer review and any attached files.

Reviewer #1: **Yes: **r_chhokar

Reviewer #2: **Yes: **Prof Amitav Banerjee

---

## [Author Response · Author response to Decision Letter 0]

3 Jun 2022

Dear Dr. Sonu

On behalf of my co-authors, many thanks for taking the time to review our paper and enabling us to submit a revised manuscript.

We have addressed all of the comments raised in the desision letter. Please see, the cover letter and the "Response to Reviewers" documents. 

Best regards,

Mariam Safi

---

## [Decision Letter · Decision Letter 1]

4 Jul 2022

PONE-D-22-05861R1The application of resilience assessment grid in healthcare: A scoping reviewPLOS ONE

Dear Dr. Mariam Safi

Thank you for submitting your manuscript to PLOS ONE. After careful consideration, we feel that it has merit but does not fully meet PLOS ONE’s publication criteria as it currently stands. Therefore, we invite you to submit a revised version of the manuscript that addresses the points raised during the review process.

We look forward to receiving your revised manuscript.

Kind regards,

Sonu Goel, MD

Academic Editor

PLOS ONE

Journal Requirements:

Reviewers' comments:

Reviewer's Responses to Questions

**Comments to the Author**

1. If the authors have adequately addressed your comments raised in a previous round of review and you feel that this manuscript is now acceptable for publication, you may indicate that here to bypass the “Comments to the Author” section, enter your conflict of interest statement in the “Confidential to Editor” section, and submit your "Accept" recommendation.

Reviewer #1: All comments have been addressed

2. Is the manuscript technically sound, and do the data support the conclusions?

Reviewer #1: Partly

3. Has the statistical analysis been performed appropriately and rigorously? 

Reviewer #1: I Don't Know

4. Have the authors made all data underlying the findings in their manuscript fully available?

Reviewer #1: Yes

5. Is the manuscript presented in an intelligible fashion and written in standard English?

Reviewer #1: Yes

6. Review Comments to the Author

Reviewer #1: Some below mentioned minor corrections are required:

Line 24:Following sentence can be reframed as: This scoping review aims to understand the practical application of RAG method and its outcomes in healthcare.

Line 33: add "the": met the inclusion criteria.

reframe the next sentence, in 2 sentences: Diversities were found across study designs (instead of design) and methodologies(instead of methods). Qualitative designs(instead of methods)and literature reviews...practice.

Line 35:replace "apply" with applied it in practice.

Line 42- omit "better"

Line 43: replace " to using it to" with "and use it to"....initiatives.

Line 56: replace "sets" with "set"

Line 82, 83,84: use abbreviation- RAG

Line 97:reframe this sentence-"The inclusion criteria was to select studies from.....reports".

Line 120 It should be- "as per advice of the...librarian".

Line 127: it should be-"after removal of the duplicates in Endnote,...."

Line 155: reframe this sentence as -"after removal of duplicates,1365 citations were retained".

Line 156: reframe this as- "After the full text assessment, 12 articles were included and 31 articles were excluded.

comment-we should always mention inclusions first .

Line185: add "onwards"-from 2018 onwards.

Line 188- it should be- "in" intensive care unit instead of an.

Line 189: it should be-"Eight studies employed.....".instead of eight of the studies....

Line 191: reframe sentence as-"All studies employed a case study design except one study(14) which had cross sectional design".

Line194: it should be-"Eleven studies were prospective".

Line 265: pls add" items" in the end of sentence after 3.

Line 280- it should be- one of the study instead of studies.

Line 336: reframe it as-"A solution to this can be.....again".

Line 338: it should be: "were" instead of "are" heterogenous

Line 355-357: Its a very long sentence. It can be reframed as- "While all organizations (plural)..... many organizations(plural).....................to the activities(plural)". "However not all........resilient".

Line 379:replace" is" with "are".

Line 380: unclear sentence to me after "and". Please reframe it for better understanding.

Line 412-"better " can be omitted

7. PLOS authors have the option to publish the peer review history of their article (what does this mean?). If published, this will include your full peer review and any attached files.

Reviewer #1: **Yes: **Reshmi Chhokar

---

## [Author Response · Author response to Decision Letter 1]

14 Jul 2022

Please see the attached letter "Response to Reviewers".

---

## [Decision Letter · Decision Letter 2]

21 Sep 2022

PONE-D-22-05861R2The application of resilience assessment grid in healthcare: A scoping reviewPLOS ONE

Dear Dr. Safi,

Thank you for submitting your manuscript to PLOS ONE. After careful consideration, we feel that it has merit but does not fully meet PLOS ONE’s publication criteria as it currently stands. Therefore, we invite you to submit a revised version of the manuscript that addresses the points raised during the review process.

We look forward to receiving your revised manuscript.

Kind regards,

Yusuke Tsutsumi

Academic Editor

PLOS ONE

Journal Requirements:

Reviewers' comments:

Reviewer's Responses to Questions

**Comments to the Author**

1. If the authors have adequately addressed your comments raised in a previous round of review and you feel that this manuscript is now acceptable for publication, you may indicate that here to bypass the “Comments to the Author” section, enter your conflict of interest statement in the “Confidential to Editor” section, and submit your "Accept" recommendation.

Reviewer #1: All comments have been addressed

2. Is the manuscript technically sound, and do the data support the conclusions?

Reviewer #1: Yes

3. Has the statistical analysis been performed appropriately and rigorously? 

Reviewer #1: I Don't Know

4. Have the authors made all data underlying the findings in their manuscript fully available?

Reviewer #1: Yes

5. Is the manuscript presented in an intelligible fashion and written in standard English?

Reviewer #1: Yes

6. Review Comments to the Author

Reviewer #1: General comment-please leave a space after putting reference number brackets, follow the same throughout the article for uniformity.

Line59-it should be- add are-"original questions that are based on....."

Line 65- replace "that" with "which" to make sentence sound better. "which recognises that......"

Line 75- replace academics with academicians

Line 100- sentence should be better framed-"The articles were excluded...." as- "The articles that described resilience assessment grids in non-healthcare settings, resilience papers not using RAG, individual or community resilience, conference abstracts with no full texts, or were theoretical/review papers were excluded.

Line 101-what is meant by individual or community resilience, it looks incomplete to me. Please reframe it.

Line 136- it should be the excel instead of Excel

Line 143-"The second and the third reviewer..." it should be.

Line 235- It should be allocation instead of allocations

Line 253-it should be -"psychological safety and problem solving".

Line 286- it should be anesthesia domain or department or unit. Please check original article, and replace with suitable option.

Line 301- replace "overview over.....".with "overview about...."

Line 355- replace "many organization..." with "many organizations..."

Line 389- it can be better framed as-"healthcare still evolving, evidenced by........2018".

Line 390- omit and after however

Line 413-omit RAG with "it"-as its repeating

7. PLOS authors have the option to publish the peer review history of their article (what does this mean?). If published, this will include your full peer review and any attached files.

Reviewer #1: **Yes: **Reshmi Chhokar

---

## [Author Response · Author response to Decision Letter 2]

27 Sep 2022

Please see the "Response to Reviewers" document.

---

## [Decision Letter · Decision Letter 3]

21 Oct 2022

PONE-D-22-05861R3The application of resilience assessment grid in healthcare: A scoping reviewPLOS ONE

Dear Dr. Safi,

Thank you for submitting your manuscript to PLOS ONE. After careful consideration, we feel that it has merit but does not fully meet PLOS ONE’s publication criteria as it currently stands. Therefore, we invite you to submit a revised version of the manuscript that addresses the points raised during the review process.

We look forward to receiving your revised manuscript.

Kind regards,

Yusuke Tsutsumi

Academic Editor

PLOS ONE

Journal Requirements:

Reviewers' comments:

Reviewer's Responses to Questions

**Comments to the Author**

1. If the authors have adequately addressed your comments raised in a previous round of review and you feel that this manuscript is now acceptable for publication, you may indicate that here to bypass the “Comments to the Author” section, enter your conflict of interest statement in the “Confidential to Editor” section, and submit your "Accept" recommendation.

Reviewer #1: All comments have been addressed

2. Is the manuscript technically sound, and do the data support the conclusions?

Reviewer #1: Yes

3. Has the statistical analysis been performed appropriately and rigorously? 

Reviewer #1: I Don't Know

4. Have the authors made all data underlying the findings in their manuscript fully available?

Reviewer #1: Yes

5. Is the manuscript presented in an intelligible fashion and written in standard English?

Reviewer #1: Yes

6. Review Comments to the Author

Reviewer #1: Line 255-256: we can omit Chuang et.al from the sentence "In the first study.....an expert panel", as the sentence prior to it mentions that Chuang et .al did two related studies.

Line 260: pls put space after the bracket closes "(SNA) to"

7. PLOS authors have the option to publish the peer review history of their article (what does this mean?). If published, this will include your full peer review and any attached files.

Reviewer #1: **Yes: **Reshmi Chhokar

---

## [Author Response · Author response to Decision Letter 3]

23 Oct 2022

please see the attached documents.

---

## [Editor Report · Decision Letter 4]

25 Oct 2022

The application of resilience assessment grid in healthcare: A scoping review

PONE-D-22-05861R4

Dear Dr. Safi,

We’re pleased to inform you that your manuscript has been judged scientifically suitable for publication and will be formally accepted for publication once it meets all outstanding technical requirements.

Kind regards,

Yusuke Tsutsumi

Academic Editor

PLOS ONE